# Fabrication of Substrate-Integrated Waveguide Using Micromachining of Photoetchable Glass Substrate for 5G Millimeter-Wave Applications

**DOI:** 10.3390/mi14020288

**Published:** 2023-01-22

**Authors:** Seung-Han Chung, Jae-Hyun Shin, Yong-Kweon Kim, Chang-Wook Baek

**Affiliations:** 1Department of Electrical and Computer Engineering, Seoul National University, Seoul 08826, Republic of Korea; 2School of Electrical and Electronics Engineering, Chung-Ang University, Seoul 06974, Republic of Korea

**Keywords:** millimeter-wave, substrate integrated waveguide, micromachining, photoetchable glass, through glass via

## Abstract

A millimeter-wave substrate-integrated waveguide (SIW) was firstly demonstrated using the micromachining of photoetchable glass (PEG) for 5G applications. A PEG substrate was used as a dielectric material of the SIW, and its photoetchable properties were used to fabricate through glass via (TGV) holes. Instead of the conventional metallic through glass via (TGV) array structures that are typically used for the SIW, two continuous empty TGV holes with metallized sidewalls connecting the top metal layer to the bottom ground plane were used as waveguide walls. The proposed TGV walls were fabricated by using optical exposure, heat development and anisotropic HF (hydrofluoric acid) etching of the PEG substrate, followed by a metal sputtering technique. The SIW was fed by microstrip lines connected to the waveguide through tapered microstrip-to-waveguide transitions. The top metal layer, including these feedlines and transitions, was fabricated by selective metal sputtering through a silicon shadow mask, which was prefabricated by a silicon deep-reactive ion-etching (DRIE) technique. The developed PEG-based process provides a relatively simple, wafer-level manufacturing method to fabricate the SIW in a low-cost glass dielectric substrate, without the formation of individual of TGV holes, complex time-consuming TGV filling processes and repeated photolithographic steps. The fabricated SIW had a dimension of 6 × 10 × 0.42 mm^3^ and showed an average insertion loss of 2.53 ± 0.55 dB in the Ka-band frequency range from 26.5 GHz to 40 GHz, with a return loss better than 13.86 dB. The proposed process could be used not only for SIW-based devices, but also for various millimeter-wave applications where a glass substrate with TGV structures is required.

## 1. Introduction

The advent of 5G communications has been accelerating the development of radio frequency (RF) devices and components working at millimeter-wave frequencies. For many millimeter-wave applications, such as wireless communications, automotive radars, high-resolution imaging systems and gigabit backbone networks, demand for small-footprint, high-density, low-loss and low-cost system integration is continuously increasing [1,2]. Packaging substrates or interposers with through substrate via structures are widely used to demonstrate 2.5D/3D packaging schemes with the fabricated devices [3,4,5,6]. In the choice of the substrate or interposer material, various factors, such as an electrical loss, precision machineability, processing yield and mechanical stability, should be considered. Conventional dielectric substrates with a low dissipation factor (tanδ) used for millimeter-wave applications include traditional printed circuit boards (PCBs) and low-temperature co-fired ceramics (LTCCs) [7], as well as organic substrates, such as liquid crystal polymers (LCPs) [8] and polytetrafluoroethylenes (PTFEs) [9]. LTCC is a robust packaging substrate for multilayer circuit integration with an ease of via formation, but the feature sizes are large and the fabrication cost is relatively high. Organic substrates have good electrical performances and low-cost manufacturing infrastructures, but dimensional tolerances, surface roughness and substrate warpage are the major issues to be addressed.

As an alternative to resolve these challenges, glass substrate with through glass vias (TGVs) has recently been used for the packaging substrate or interposers for high-frequency applications [10,11,12,13,14,15,16,17,18,19,20,21,22,23,24,25,26]. Glass, as a substrate material, is emerging because of its low electrical loss, ability to form fine-pitch metal lines and spaces, machineability to create micro-size TGVs, high dimensional stability, a closely matched coefficient of thermal expansion (CTE) to silicon dies, and productivity in thin and large panels. Integrated RF passives based on the glass interposers with TGVs, such as filters [12,13,20], 3D inductors [13,14], substrate-integrated waveguides [16,17,21,22] and antennas [18,23,24,25,26], have been reported. In these applications, the formation of TGV holes with fine sizes and pitches in the glass interposers and their metallization processes are crucial factors. Conventional methods to form TGV holes, such as drilling and sandblasting, or micromachining processes, such as wet or dry etching, laser pulse ablation and electro discharging techniques, are not usually compatible with wafer-level batch fabrication processes and have limitations in creating very small, fine-pitched and high-aspect-ratio empty holes in glass. In addition, it is another challenge to obtain void-free metal vias filled in the TGV holes using a classical electroplating process [16]. Glass interposers with tungsten-coated through glass silicon via (TGSV) structures by combining silicon deep-reactive ion-etching (DRIE), selective metal coating on the silicon vias and glass reflow technique, were demonstrated for millimeter-wave applications by our group [21,25,26]. This platform provides rigid, completely filled, void-free TGV structures showing comparable electrical performances to the metal vias. The whole fabrication process, however, is still somewhat complex, expensive and time-consuming.

Photoetchable glass (PEG) is a special type of glass showing a large anisotropy of etch rates when exposed to ultraviolet (UV) light and heat treatment [27]. The photosensitivity and etchability of the exposed region in a hydrofluoric acid (HF) can make wafer-level fabrication of very small sized, high-aspect-ratio TGV holes in the glass substrate very easy, which is hard to be achieved in conventional drilling or machining techniques. Thus, PEG could be utilized as a substrate to demonstrate interposers and integrated RF passives. However, very few works on the RF applications of PEG substrates have been reported. Spiral TGV inductors and lumped filters at 2.1 GHz [14], and microstrip patch antenna with TGVs at 67 GHz [15], are very rare examples. In this paper, photoetchable glass was investigated as a potential substrate material for millimeter-wave applications. In our work, we demonstrated a substrate-integrated waveguide (SIW) operating at the Ka-band in a PEG substrate, by using the wafer-level micromachining processes of PEG. SIWs are emerging as low-profile waveguides that can be integrated into a dielectric substrate while still showing low-loss, excellent power handling and immunity from radiation [28,29]. For these reasons, SIW technology has been widely applied to various components, such as filters, integrated antennas and other passives for millimeter-wave and terahertz applications [28,29,30,31,32,33,34,35,36]. Here, a PEG substrate was used as a dielectric material of the SIW, and its photoetchable properties were used to fabricate through glass via (TGV) holes. The sidewalls of the SIW connecting the top metal and bottom ground layers were realized by forming two continuous empty TGV holes in the PEG substrate and then metallizing them using sputtering processes through a prefabricated silicon shadow mask. The proposed fabrication process enables relatively simple, wafer-level manufacturing of the SIW that has precise dimensions required for millimeter-wave devices, in low-cost glass substrates, without the use of additional photolithographic steps onto the wafer with TGV holes and time-consuming complete metal-filling processes into those TGV holes. The electrical performances of the fabricated SIWs were experimentally measured and compared with those of other glass-based SIWs.

## 2. Design and Simulation

### 2.1. Dielectric Properties of PEG Substrate

PEGs, which are normally known by their trademark names, such as FORTURAN (Schott, Germany) or APEX (Life Bioscience Inc., Albuquerque, NM, USA), are lithium aluminum silicates with some well-defined impurities of metal oxides that significantly contribute to photosensitive properties. These impurities include photosensitive metals, with silver (Ag) being used more than others, and optical sensitizers, among which cerium (Ce) is the most important. Typical procedures to fabricate microstructures in PEG are described in the literature [37,38]. The PEG substrate is first exposed to mid-to-near UV light whose wavelength ranges from 260 to 360 nm through a photomask, which is normally composed of quartz. Ce^3+^ ions absorb photons and release one electron to enter the stable Ce^4+^ state, and this electron is absorbed by Ag^+^ ions reduced to Ag atoms. The exposed glass is then heated to higher temperatures, typically between 580~680 °C. During this “heat development” process, Ag atoms are clustered into bigger nuclei around which the glass crystallizes to form lithium metasilicate (Li_2_SiO_3_), producing the latent image of the photomask [39]. The exposed region can be etched away, typically in a solution of 10% hydrofluoric acid (HF), because the etching ratio between the crystallized parts and bulk part is known to be more than 20:1 [27].

To design the SIW using PEG, the electrical properties of the PEG substrate need to be evaluated. In our study, the dielectric constant and loss tangent of the PEG substrate (MEG2; MicroFab, Seoul, Republic of Korea) at 28 GHz were experimentally measured using a split cylinder resonator method [40], where the cylindrical resonator is split in the middle and a plate sample is placed between the two. The permittivity is obtained from the change in frequency and the dielectric loss is obtained from the change in *Q* values. The split cylinder uses the TE_011_ mode of resonance, where the electric flux circulates in the sample plane, eliminating the negative effects of ends and allowing accurate measurements. PEG plate samples were fabricated by dicing the substrate into a rectangular plate of 30 mm × 50 mm, followed by polishing them down to the thickness of 154 μm. Two different samples were prepared, one of which is an original unprocessed PEG, while the other is thermally treated at a temperature of 585 °C for 2 h to emulate the heat development conditions. The prepared samples were loaded in the split cylinder resonator (CR-728; EMLabs, Kobe, Japan) and measured using a vector network analyzer (N5291A; Keysight Technologies, Santa Rosa, CA, USA). The measured results are summarized in Table 1. The dielectric constant slightly decreased from 6.1 to 5.8 after heat development, but the loss tangent values were almost the same for both samples. In the design of the SIW, therefore, a dielectric constant of 5.8 and a loss tangent of 0.014 were used. The measured values are similar to the reported value of PEG at 60 GHz, which was a dielectric constant of 6.4 and a loss tangent of 0.022 [15].

### 2.2. Design and Simulation of SIW

In our work, SIW was selected to demonstrate the potential of PEG in the millimeter-wave regimes. The entire three-dimensional schematic view of the proposed SIW is illustrated in Figure 1. The SIW is composed of a PEG dielectric substrate, TGVs, top metal patterns and bottom ground metal layer. Instead of forming the cylindrical metal-filled via arrays normally used for SIWs, two continuous rectangular empty TGV holes with their vertical sidewalls coated with a thin metal layer were utilized. These metal-coated TGV holes behave like the conductor walls of a conventional 3D rectangular waveguide. Therefore, leakage loss through the gap between the via arrays of the conventional SIW is minimized. The TGV holes can be fabricated in a waver level using the photoetchable properties of PEG, by following the procedures briefly described in Section 2.1. Instead of filling the TGV holes using photolithography and electroplating processes, only the sidewalls of the TGV holes are coated with thin metal layers during the consecutive two sputtering processes: one selective sputtering for the top metal patterns (including microstrip feedlines and tapered transitions) using a silicon shadow mask fabricated by silicon deep-reactive ion-etching (DRIE), and another, final sputtering without any shadow mask for the bottom ground metals. The proposed process does not require additional photolithographic processes to pattern the top metal layers, which is difficult to perform on the wafer with TGV holes. Furthermore, the via-filling process using electroplating—which is also difficult and takes a long time to obtain completely filled, void-free metallic vias for a substrate with a thickness of hundreds of micrometers—is also not necessary. The precision of the top metal patterns, which is important to fabricate millimeter-wave devices that usually have small sizes, could be determined by the accuracy of the silicon DRIE process, which is normally a few micrometers. In addition, the fabricated silicon shadow mask can be repeatedly used for the process of other substrates. These characteristics of the proposed process could make wafer-level fabrication process of the SIW relatively simpler than the previous works using TGV structures.

When the width of the waveguide is larger than its thickness, the cutoff frequency of the dominant TE_10_ mode is given by the following well-known equations [41]:(1)$$f_{c,10}=\frac{c}{2w\sqrt{\varepsilon_{r}}}$$
where $f_{c,10}$ is the cutoff frequency of TE_10_ mode, w is the width of the waveguide, $\varepsilon_{r}$ is the dielectric constant of the substrate and c is the speed of light in free space. In our study, an SIW with a cutoff frequency of 20.7 GHz was designed and optimized using commercial full-wave electromagnetic simulation software (Ansys HFSS, R16). The PEG substrate was supplied in the form of a 100 mm × 100 mm square plate with a thickness of 420 μm; thus, the thickness of the SIW (h) was also fixed to the same value here. A 50 Ω microstrip line was used for the feeding lines, and linearly tapered microstrip line transitions were connected between the microstrip line and the SIW to transform the quasi-TEM mode of the microstrip line into the TE_10_ mode in the waveguide [42]. Smooth transition using this tapered transformer ensures field matching between the microstrip line and the waveguide over a broad bandwidth. After determining the waveguide width, the tapered section was designed using HFSS simulation, interrelating the line width of the microstrip line with the width of the SIW. The top view of the designed SIW metal pattern on the PEG substrate is shown in Figure 2. Detailed dimensions are $w_{v}$ = 0.205 mm, $L_{v}$ = 3.6 mm, $w_{ms}$ = 0.64 mm, $L_{ms}$ = 1.6 mm, $w_{t}$ = 1.2 mm, $L_{t}$ = 1.5 mm, L = 3.95 mm and w = 3.0 mm, respectively.

The simulated S-parameters of the designed SIW are shown in Figure 3. The average insertion loss of the SIW at the Ka-band (26.5 GHz to 40 GHz), including all microstrip feedlines and transitions, was estimated to be 1.41 ± 0.13 dB (1.79 dB at 40 GHz). In the photomask designed to fabricate the silicon shadow mask, the designed SIW and microstrip line patterns were included. 

## 3. Fabrication Process

The overall fabrication process of the proposed SIW is illustrated in Figure 4. A 420 μm thick, 100 mm × 100 mm square-shaped PEG substrate was used as a dielectric substrate of the SIW. To create the TGV holes for the SIW, the glass substrate was firstly exposed by a 310 nm near-UV light source (ML-100F; Mikasa Co., Ltd., Tokyo, Japan) with a dose of 7.92 J/cm^2^ using a soda-lime glass photomask (Figure 4a). As mentioned in Section 2.1, a quartz mask that can transmit the required wavelength of 310 nm well is normally utilized. In our experiment, however, exposure conditions were optimized for a soda-lime glass mask because the quartz mask is expensive and also very sensitive to the optical dose. Then, the exposed substrate was heat developed and polished (Figure 4b). The heat development conditions were carefully managed to minimize the dimensional changes of the patterns. The substrate was first heated up in the furnace to 350 °C with a rate of 3 °C/min and maintained for 30 min. The furnace temperature was further increased up to 585 °C with a rate of 2 °C/min and maintained for 2 h, and then the substrate was naturally cooled down to the room temperature. During this heat treatment step, the exposed area became foggy due to the crystallization of glass around Ag nanoclusters. Before the etching of the exposed region, both sides of the glass substrate were polished by chemical mechanical polishing (CMP). This CMP step is very important in order to keep the surface roughness of the PEG substrate as smooth as possible during the following etching step, which, in turn, affects the conductor loss of the deposited metal layer onto the substrate. The heat-developed substrate was dipped into a 10% HF solution for 450 s to etch away the exposed TGV regions (Figure 4c). The etch rate of the glass was measured to be about 60 μm/min.

After HF etching, both sides of the PEG substrate are metallized (Figure 4d). First, top metal patterns, including waveguide, transitions and microstrip feedlines, were formed by the selective sputtering of a Ti/Cu (10 nm/500 nm) layer through the prefabricated silicon shadow mask, which is precisely aligned with the substrate. A 4-inch, 350 μm thick single crystal silicon wafer was etched through into the shape of the SIW pattern by the DRIE of silicon using an oxide etch mask, and it was utilized as a shadow mask for metal patterning (the shadow mask fabrication process is not shown in Figure 4). During this process, the sidewalls of the TGV holes were deposited with copper to some extent. Then, the bottom surface was coated with a metal layer, which is a ground plane of the SIW, by further Ti/Cu sputtering without any shadow mask. During these two consecutive sputtering processes, the sidewalls of the TGV holes were twice-covered with the deposited metal. Good electrical connections between the top and bottom metal layers were made thanks to the excellent step coverage of the sputtering process. Finally, the substrate was diced out into a 6 mm × 10 mm sample for measurement (Figure 4e).

The PEG substrate exposed to the UV light and heat developed in the furnace is shown in Figure 5a. As described, the exposed region became opaque due to the crystallization effect. After polishing both sides of the substrate and etching it in a 10% HF solution, TGV holes were successfully fabricated, as shown in Figure 5b. The fabricated silicon shadow mask for the following selective metal deposition is shown in Figure 5c. The top and bottom views of the fabricated SIW sample after dicing are shown in Figure 5d.

## 4. Experimental Results and Discussion

Before measuring the RF performances of the fabricated SIW, the dimensions of each part of the sample were measured and compared with the designed values, as shown in Table 2. The fabricated dimensions tended to be smaller than the designed values, which might be caused by various factors, such as the resolution of the PEG process, an alignment error between the shadow mask and PEG substrate, warpage of the silicon or PEG substrate, inaccuracy or nonuniformity of the DRIE-etched patterns in the silicon shadow mask, etc. It can be seen that the parameters $w_{v}$, $L_{v}$ and w, which are directly related to the resolution of the PEG patterning process, do not deviate much from the design. However, the other parameters, specifically depending on the size of the shadow mask patterns and physical state of the substrates, showed relatively larger disparities from the designed values, with a maximum deviation of up to 40 μm. The effect of these dimensional variations on the RF performances, thus, cannot be completely ignored, especially for the case of millimeter-wave devices where high-dimensional accuracies are required. Therefore, EM simulation of the SIW was performed again using these measured dimensions, and the results were compared with the measurement results.

The RF performances of the fabricated SIW were measured using a commercially available universal test fixture (3680V; Anritsu Corp., Atsugi, Japan) and a vector network analyzer (N5227B; Keysight Technologies, Santa Rosa, CA, USA). The experimental setup is shown in Figure 6. A standard SOLT (Short-Open-Load-Thru) calibration process was performed with a commercial calibration kit (36804B-10M; Anritsu Corp., Atsugi, Japan) for calibration.

The measured S-parameters of the fabricated SIW were compared with the simulation results in Figure 7. It can be seen that the simulated average insertion loss increased to 2.05 ± 0.24 dB (2.65 dB at 40 GHz) when the dimensional changes from the original design were reflected in the simulation. The measured S-parameters, as expected and shown in the figure, agree well with these new simulation results, including these fabrication errors. The measured average insertion loss of the fabricated SIW, however, saw a somewhat higher value of 2.53 ± 0.55 dB (2.94 dB at 40 GHz), mainly due to the increase in measured loss above 32 GHz. For the purpose of comparison, the measured S-parameters of a 10 mm long, 50 Ω microstrip line fabricated on the same PEG substrate is shown in Figure 8. In the Ka-band, the insertion loss of the microstrip line was measured to be 2.26 ± 0.37 dB (2.49 dB at 40 GHz).

The measured insertion loss of the fabricated PEG SIW is higher than that of the reported SIWs fabricated on the borosilicate glass [16,21,22]. One main reason for this is the higher dissipation factor (0.016) of PEG, which is an order of magnitude higher than the borosilicate glass (0.0037 at 1 MHz [16,21], 0.006 at 24–40 GHz [22]). Another factor that contributes to an increase in the insertion loss is the roughness of the conductor metals, which, in turn, is strongly dependent on the surface roughness of the PEG substrate or sidewalls of TGV holes in the PEG. In Figure 9, we compared the AFM (atomic force microscope) images of unexposed surfaces for two PEG substrates: one etched to form TGV holes right after the heat development, and the other etched after being polished by CMP. The surface roughness of the PEG directly etched in a 10 wt% HF solution without performing a CMP process was measured to be about 15 times higher than the polished PEG substrate. This increase in surface roughness in the unexposed region is actually observed at the moment when heat development is performed. In addition, we can observe that the sidewalls of the TGV holes etched in HF are very rough compared with the polished top surface, as shown in the SEM (scanning electron microscope) image of Figure 10. In our process, the roughened sidewall of the TGV holes is directly reflected in the thin layers of the sputtered metal, which are used as via conductors instead of filled-up metals.

To confirm the effect of surface roughness on the RF performances, the S-parameters of the two SIWs fabricated on the PEG substrate with or without the CMP process are compared in Figure 11. As shown in the figure, insertion loss at 40 GHz was increased from 2.94 dB to 4.32 dB when the surface of the PEG substrate was not polished by CMP. Therefore, the CMP process prior to the etching of the exposed region needed to be performed, as described in Section 3, to improve insertion losses. 

As mentioned before, very few works about microwave applications of PEG are reported, especially in the regime of millimeter-wave frequencies. Since we could not find the SIW realized in the PEG substrates, the performances of our fabricated SIW on the PEG substrate were compared with some other glass-based SIWs, including our previous results, which were mostly manufactured on the borosilicate glass substrates, as presented in Table 3. Due to the higher dielectric loss factor described above, the insertion loss was measured to be somewhat higher than that of the SIWs on the borosilicate glass. In our work, a 10 mm long microstrip line on the PEG substrate showed an insertion loss of 2.49 dB at 40 GHz. In the previous reference, coplanar waveguides (CPWs) and grounded coplanar waveguides (GCPWs) fabricated on the PEG substrate exhibited an insertion loss of 3.3 dB/cm at 60 GHz [14].

## 5. Conclusions

In this study, we proposed a manufacturing method to fabricate SIWs based on the micromachining of photoetchable glass (PEG). Empty TGV holes were fabricated by the UV exposure, heat development and HF etching of PEG, and the sidewalls of the TGV holes were metallized with selective sputtering processes using a silicon shadow mask to connect the top metal and bottom ground plane. By using this process, the wafer-level manufacturing of SIWs in a low-cost glass substrate can be carried out in a relatively simple way without further photolithographic steps or time-consuming via metallization processes. An SIW operating at the Ka-band was designed and fabricated by the developed process. The measured average insertion loss of the SIW was 2.53 ± 0.55 dB with a return loss better than 13.86 dB, from 26.5 to 40 GHz. A microstrip line fabricated on the PEG substrate showed an average insertion loss of 2.26 ± 0.37 dB in the same frequency range. Due to the higher dielectric loss factor of PEG, the fabricated SIW on the PEG substrate tends to show a somewhat higher insertion loss than the SIW on the borosilicate glass. The PEG substrate, however, still has the potential to be applied to millimeter-wave passives or packages for 5G applications because of its simple, wafer-level machineability to fabricate fine-pitched micro TGVs in the glass dielectric material. The process developed in this work could be applied to demonstrate millimeter-wave devices such as SIW-based resonators, filters, 3D inductors and helical antennas, or interposers for integrated packaging where conductive TGVs are required.

## Figures and Tables

**Figure 1 micromachines-14-00288-f001:**
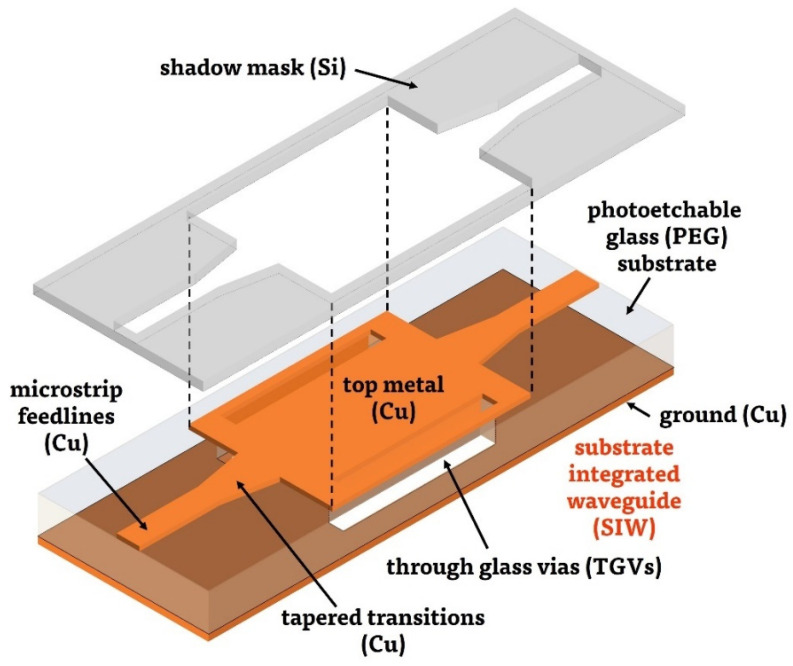
Three-dimensional schematic view of the proposed PEG-based SIW (including silicon shadow mask for patterning top metal layers).

**Figure 2 micromachines-14-00288-f002:**
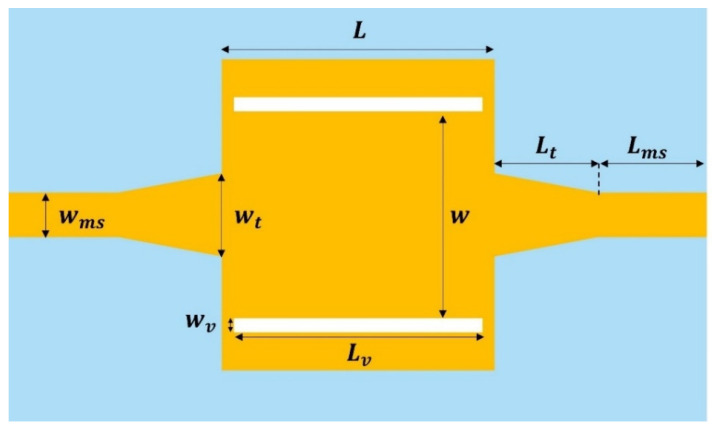
Top view of the SIW with the design parameters.

**Figure 3 micromachines-14-00288-f003:**
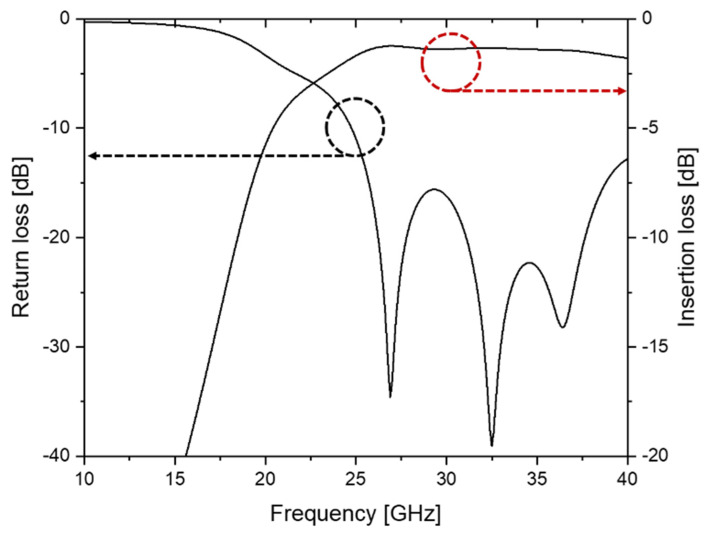
Simulated S-parameters of the designed SIW.

**Figure 4 micromachines-14-00288-f004:**
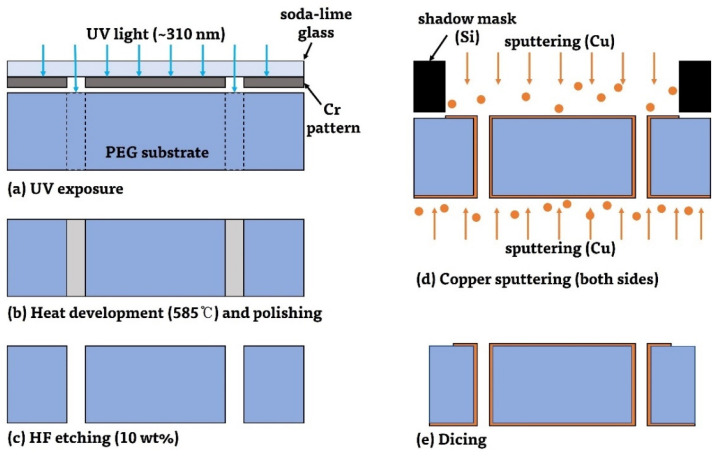
The overall fabrication process of the proposed SIW. Here, the fabrication process of the silicon shadow mask is not illustrated separately.

**Figure 5 micromachines-14-00288-f005:**
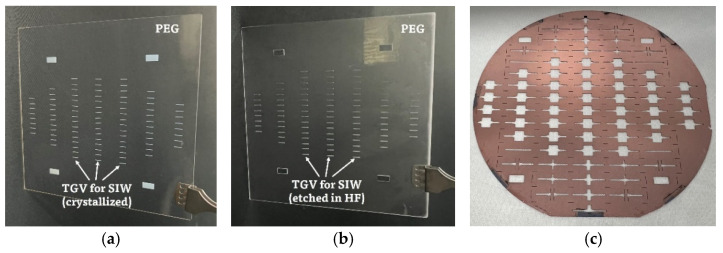
(**a**) PEG substrate after exposure and heat development. (**b**) PEG substrate after HF etching of TGV holes. (**c**) Si shadow mask for patterning of top metal layers of SIWs (after utilized for copper sputtering). (**d**) Top and bottom side views of the fabricated SIW samples.

**Figure 6 micromachines-14-00288-f006:**
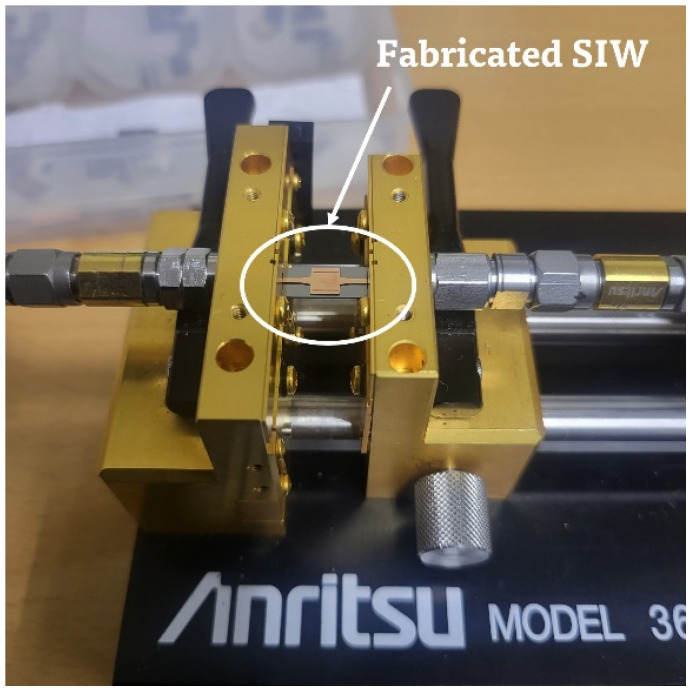
Experimental setup for measuring S-parameter of the fabricated SIW.

**Figure 7 micromachines-14-00288-f007:**
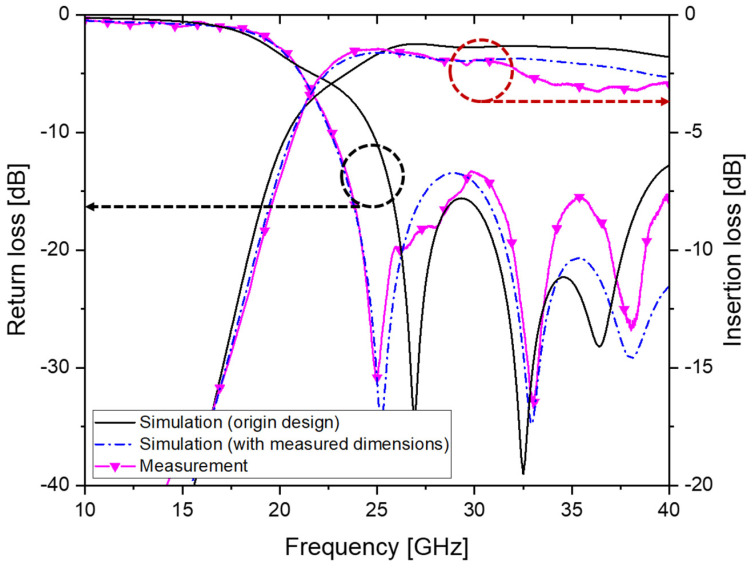
Measured S-parameters of the fabricated SIW compared with the simulation results.

**Figure 8 micromachines-14-00288-f008:**
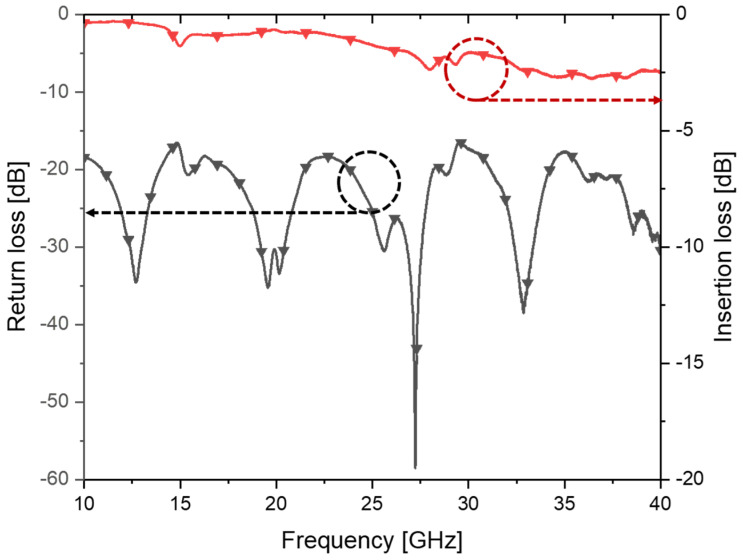
Measured S-parameters of the fabricated 10 mm long microstrip line.

**Figure 9 micromachines-14-00288-f009:**
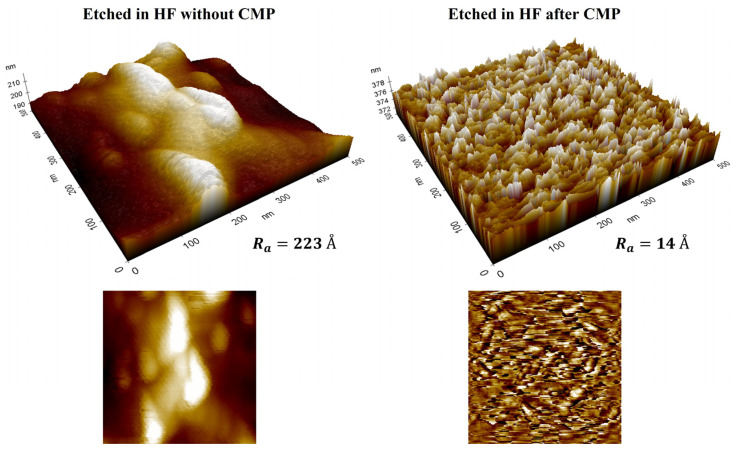
AFM images of the surface of the PEG substrate etched in HF without CMP (**left**) and after CMP (**right**).

**Figure 10 micromachines-14-00288-f010:**
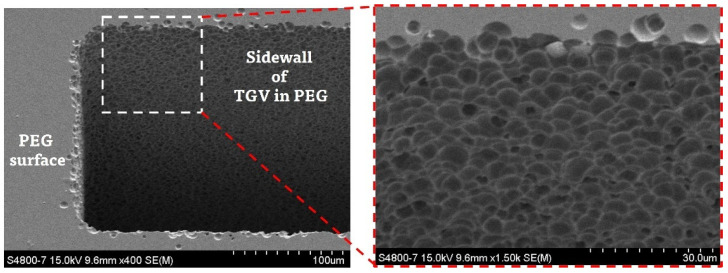
SEM images of the TGV holes in the PEG substrate.

**Figure 11 micromachines-14-00288-f011:**
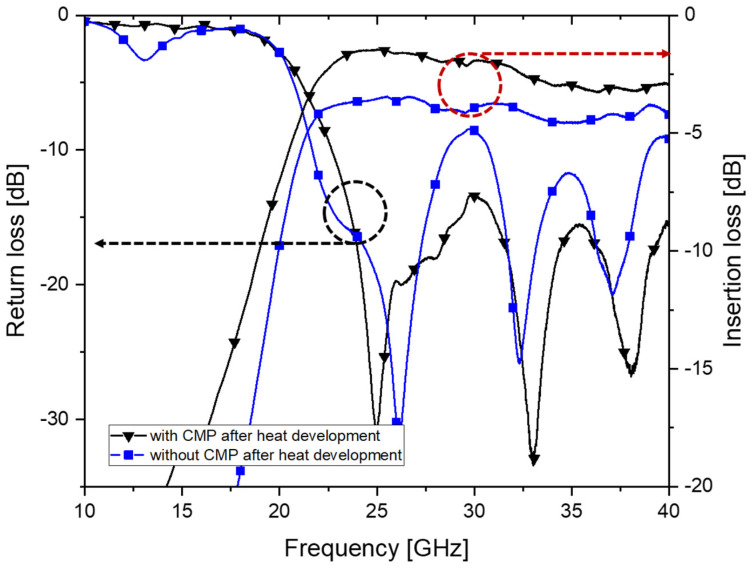
The measured S-parameter of the fabricated SIWs with and without performing CMP process after heat development.

**Table 1 micromachines-14-00288-t001:** Measured permittivity and loss tangent values of the PEG samples at 28 GHz.

Samples	Dielectric Constant	Loss Tangent
PEG (original)	6.1	0.014
PEG (heat developed)	5.8	0.014

**Table 2 micromachines-14-00288-t002:** Comparison of the dimensional parameters of the SIW.

Parameters	Design	Measurement	Description
$w_{v}$	0.205 mm	0.202 mm	Width of the TGV wall
$L_{v}$	3.6 mm	3.591 mm	Length of the TGV wall
$w_{ms}$	0.64 mm	0.629 mm	Width of the microstrip line
$L_{ms}$	1.6 mm	1.58 mm	Length of the microstrip line
$w_{t}$	1.2 mm	1.18 mm	Width of the tapered transition
$L_{t}$	1.5 mm	1.48 mm	Length of the tapered transition
L	3.95 mm	3.91 mm	Length of the SIW
w	3.0 mm	3.005 mm	Width between the two TGV walls
h	0.420 mm	0.415 mm	Thickness of the PEG substrate

**Table 3 micromachines-14-00288-t003:** Performances of the SIWs fabricated on the glass substrate.

Ref.	$\mathrm{Substrate}\;\mathrm{Thickness}/\varepsilon_{r}$ /tan δ	Vias	Frequency [GHz]	Device Length [mm]	Insertion Loss [dB]
[16]	Borosilicate glass (350 μm/4.6/0.0037 ^1^)	Electroplated Cu	20–45	10.0 ^2^	<1.4 dB ^2^
[21]	Borosilicate glass (350 μm/4.6/0.0037 ^1^)	Tungsten-coated Si	20–45	7.0 ^2^	<1.15 dB ^2^
[22]	Borosilicate glass with low-loss polymer lamination (100 μm/5.4/0.006)	Electroless-plated Cu + SAP (semi-additive patterning)	24–40	4.0 ^3^	0.64 dB ^3,4^
This work	Photoetchable glass (420 μm/5.81/0.014)	Sputtered Cu on the TGV walls	26.5–40	10.0 ^2^	2.53 ± 0.55 dB ^2^

^1^ Values at 1 MHz, which is not measured directly at the target frequency. ^2^ Values including all the transitions and feeding lines. ^3^ Values of SIW only. ^4^ De-embedded value.

## Data Availability

The data presented in this study are available on request from the corresponding author. The data are not publicly available due to privacy restrictions.

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
