# Peer review of "Fabrication of Substrate-Integrated Waveguide Using Micromachining of Photoetchable Glass Substrate for 5G Millimeter-Wave Applications"

_micromachines, 2023, doi:10.3390/mi14020288_

Round 1

Reviewer 1 Report

Authors in this research paper have investigated and realized a manufacturing method to fabricate substrate integrated waveguides based on micromachining of photoetchable glass. TVG holes were fabricated by UV exposure and heat development of photoetchable glass, and the sidewalls of TGV holes were metallized with selective sputtering processes using a silicon shadow mask. By using this process, low-cost wafer-level manufacturing of the substrate integrated waveguide is possible without further photolithographic or metal etching processes. The idea and concept of the manuscript are interesting and they have been supported by achieving promising results. But, before final recommendation authors are requested to carefully address the following comments to improve its quality.

1) Applications of the proposed work can be added in the title of the manuscript.

2) Abstract section is short and authors can improve this part by a proper extension by addressing the following points:

a) Please briefly discuss the design process of the proposed millimeter-wave substrate integrated waveguide.

b) Please add more numerical achievements to this part.

c) Advantages and practical application(s) of the proposed SIW can be highlighted in this part.

3) Introduction section can be improved by adding more discussions and references on various applications of the SIW technology such as antennas for 5G applications. Below are helpful suggestions.

"MTM- and SIW-Inspired Bowtie Antenna Loaded with AMC for 5G mm-Wave Applications" International Journal of Antennas and Propagation, Volume 2021, Article ID 6658819, 7 pages https://doi.org/10.1155/2021/6658819.

"High-Isolation Antenna Array Using SIW and Realized with a Graphene Layer for Sub-Terahertz Wireless Applications", Scientific Reports, 11, 10218, 2021.

"On-Chip Antenna Design Using the Concepts of Metamaterial and SIW Principles Applicable to Terahertz Integrated Circuits Operating over 0.6–0.622 THz" International Journal of Antennas and Propagation, Volume 2020, Article ID 6653095, 9 pages, https://doi.org/10.1155/2020/6653095.

“High-Isolation Leaky-Wave Array Antenna Based on CRLH Metamaterial Implemented on SIW with ±30o Frequency Beam-Scanning Capability at Millimeter-Waves”, Electronics, 2019, 8, 642,15 pages.

"Antenna on Chip (AoC) Design Using Metasurface and SIW Technologies for THz Wireless Applications", Electronics, 2021, 10(9), 1120.

4) Fig.1 shows the three-dimensional schematic view of the proposed PEG-based SIW, please discuss its design process in more depth. Also, please explain its feeding mechanism.

5) Top view of the SIW with the design parameters is exhibited in Fig.2, please explain why authors have used two rectangular slots on the top side? Their performance parameters should be elaborated in more detail.

6) Why have authors used PEG substrates?

7) Practical applications of the proposed SIW implemented on PEG substrate should be discussed in depth.

8) Before concluding the manuscript, please highlight the advantages of the proposed SIW by comparing this work with some prior art. The results can be summarized in a table.

9) Please support the conclusion by adding more numerical results.

10) Reference part can be improved by a proper extension as per above mentioned suggestions.

11) Please define “TVG” in the manuscript. 

Author Response

We would like to thank for your valuable comments on our manuscript. Our detailed responses to your comments have been prepared into a separate file and attached.

Reviewer 2 Report

A comparison or at least some references of other photoetchable glasses used in microwave and millimeter-wave should contribute to the already good introduction.

The precision of the numbers in table 1 should be corrected to the tolerances of the measuring system. Six decimal points does not seem correct. 

How the resistance between the top and bottom layers was measured to be 0.24 Ω?

The authors mention some reasons of why the SIW has more losses than other works with borosilicate glass. What is the final copper thickness of the top and bottom copper layers? And what is the thickness of the copper in the vias. How do they affect the losses?  

What is the resolution of the top layer fabrication? 

Author Response

(The authors gave the same response as above.)

Reviewer 3 Report

In this paper, authors have presented the fabrication of mm-Wave substrate integrated waveguide using photoetchable dielectric glass substrate. The manuscript presents interesting scientific work. However, consider my following comments/suggestions:

1. Please show how the fabrication process of such photoetchable glass substrate is simple, low-cost, and non-time-consuming comparable to other TGVs.

2. The sentence in the introduction, “In this work, we designed and fabricated a substrate integrated waveguide (SIW) operating at Ka-band as a test vehicle by developing PEG-based microfabrication processes” is not clear.

3. To bring out the novelty, compare the proposed SIW implementation method with others in terms of manufacturing, electrical characteristics, and size requirements.

4. Add experiments photographs for the dielectric constant and loss tangent

measurements performed for section 2.1. That will help readers.

5. Modify the insertion loss plots as they are plotted for a wide range of 0 to -50 dB. May be 0 to -5/-10 dB. The variations are not clearly visible.

6. Insertion loss of 2.53 dB at 28 GHz is large as less than 2 dB is accepted. Add using what methods the insertion be improved for the proposed dielectric substrate.

Author Response

(The authors gave the same response as above.)

Round 2

Reviewer 1 Report

The reviewers' comments have been carefully considered by authors during the revision process, which has helped to improve the quality of the manuscript. So, there are no more technical comments.